

# Artificial shelters and marine infectious disease: no detectable effect of the use of casitas to enhance juvenile *Panulirus argus* in shelter-poor habitats on a viral disease dynamics

Rebeca Candia-Zulbarán,  Patricia Briones-Fourzán,  Fernando Negrete-Soto, Cecilia Barradas-Ortiz  and  Enrique Lozano-Álvarez

Instituto de Ciencias del Mar y Limnología, Unidad Académica de Sistemas Arrecifales, Universidad Nacional Autónoma de México, Puerto Morelos, Quintana Roo, México

Corresponding author
Patricia Briones-Fourzán,
briones@cmarl.unam.mx

## ABSTRACT

Casitas, low-lying artificial shelters that mimic large crevices, are used in some fisheries for Caribbean spiny lobsters (*Panulirus argus*). These lobsters are highly gregarious and express communal defense of the shelter. Scaled-down casitas have been shown to increase survival, persistence, and foraging ranges of juveniles. Therefore, the use of casitas has been suggested to help enhance local populations of juvenile *P. argus* in Caribbean seagrass habitats, poor in natural crevice shelters, in marine protected areas. Following the emergence of Panulirus argus virus 1 (PaV1), which is lethal to juveniles of *P. argus*, concern was raised about the potential increase in PaV1 transmission with the use of casitas. It was then discovered that lobsters tend to avoid shelters harboring diseased conspecifics, a behavior which, alone or in conjunction with predatory culling of diseased lobsters, has been proposed as a mechanism reducing the spread of PaV1. However, this behavior may depend on the ecological context (i.e., availability of alternative shelter and immediacy of predation risk). We conducted an experiment in a lobster nursery area to examine the effect of the use of casitas on the dynamics of the PaV1 disease. We deployed 10 scaled-down casitas per site on five 1-ha sites over a reef lagoon (casita sites) and left five additional sites with no casitas (control sites). All sites were sampled 10 times every 3–4 months. Within each site, all lobsters found were counted, measured, and examined for clinical signs of the PaV1 disease. Mean density and size of lobsters significantly increased on casita sites relative to control sites, but overall prevalence levels remained similar. There was no relationship between lobster density and disease prevalence. Dispersion parameters ($m$ and $k$ of the negative binomial distribution) revealed that lobsters tended to avoid sharing natural crevices, but not casitas, with diseased conspecifics. These results confirm that casitas provide much needed shelter in seagrass habitats and that their large refuge area may allow distancing between healthy and diseased lobsters. On eight additional sampling times over two years, we culled all diseased lobsters observed on casita sites. During this period, disease prevalence did not decrease but rather increased and varied with site, suggesting that other factors (*e.g.*, environmental) may be influencing the disease dynamics. Using scaled-down casitas in shelter-poor habitats may help efforts

to enhance juvenile lobsters for conservation purposes, but monitoring PaV1 prevalence at least once a year during the first few years would be advisable.

# INTRODUCTION

The Caribbean spiny lobster (*Panulirus argus*) is a valuable resource that sustains numerous industrial and artisanal fisheries in the wider Caribbean region (*Wahle, Linnane & Harrington, 2020*; *Doerr, 2021*). Juveniles of *P. argus* dwell in shallow reef lagoons and embayments where marine vegetation (seagrasses and macroalgal beds) abound. Marine vegetation provides settlement habitat for postlarvae of *P. argus* and protection for the smaller juveniles; however, juvenile lobsters eventually outgrow the protection afforded by the vegetation and seek nearby crevice-type shelters before migrating as subadults to the coral reef habitats where the adults live (*Butler, Steneck & Herrnkind, 2006*; *Briones-Fourzán & Lozano-Álvarez, 2013*).

Caribbean spiny lobsters are gregarious, with multiple individuals sharing crevice-type dens. Aggregated lobsters express group defense, increasing the per capita survival (*Eggleston & Lipcius, 1992*). Because the risk of predation is high for juveniles of *P. argus*, they greatly depend on available crevice shelters for survival (*Smith & Herrnkind, 1992*; *Behringer et al., 2009*). Shallow hard-bottom habitats may abound in potential shelters for juvenile spiny lobsters in the form of large sponges, coral heads, solution holes, rocky outcrops, ledges, and crevices. In contrast, soft-bottom habitats such as seagrass meadows are typically poor in crevice shelters (*Sosa-Cordero et al., 1998*; *Briones-Fourzán & Lozano-Álvarez, 2001*; *Behringer et al., 2009*), potentially causing local demographic bottlenecks for *P. argus* (*Arce et al., 1997*; *Butler & Herrnkind, 1997*; *Caddy, 2008*).

Because lobsters will also take refuge in many types of man-made structures, several highly productive Caribbean fisheries for *P. argus* have long used low-lying, flat-topped artificial shelters called "pesqueros" in Cuba, "condos" in the Bahamas, and "casitas" in Mexico and elsewhere (*Briones-Fourzán, Lozano-Álvarez & Eggleston, 2000*; *Cruz & Phillips, 2000*; *Doerr, 2021*). Concern has been raised about casitas potentially causing overexploitation, resulting in their ban in some fisheries (*e.g.*, Florida, USA) (*Ross, Butler & Matthews, 2022*). However, overexploitation is more likely to occur where open-access fisheries conditions exist (*Caddy, 2008*; *Doerr, 2021*). In the casita-based Cuban and Mexican fisheries, which have regulations such as the use of limited and enforceable spatial access rights, casitas have increased the carrying capacities of the managed areas (*Sosa-Cordero, Liceaga-Correa & Seijo, 2008*; *Headley et al., 2017*).

From the conservation viewpoint, some authors have expressed concern that the use of casitas may alter the local benthic habitats or biological communities, or affect the lobsters themselves. However, lobsters have a broad diet that includes many types of invertebrates including clams with chemosynthetic bacteria that live in seagrass habitats (*Higgs, Newton*

& Attrill, 2016), and there has been little measurable impact of casitas on the abundance of the invertebrate fauna on which lobsters feed (*Vidal & Basurto, 2003*; *Nizinski, 2007*), or on the benthic habitats where casitas are deployed (*Ross, Butler & Matthews, 2022*). There is also no evidence that casitas negatively affect subadult and adult *P. argus* (*Gittens & Butler, 2018*). Because lobsters over a broad size range commonly occupy large commercial casitas ($\sim$2 m$^2$ in area, 15 cm in height) (*Lozano-Álvarez, Briones-Fourzán & Phillips, 1991*; *Sosa-Cordero et al., 1998*; *Candia-Zulbarán et al., 2012*), other authors have cautioned against the use of casitas in nursery habitats because they may function as ecological traps for juveniles (*Gutzler, Butler & Behringer, 2015*). However, it has been shown that shelter scaling is important for survival of lobsters and for the enhancement of juveniles of *P. argus* in shelter-poor seagrass habitats (*Eggleston et al., 1990*; *Arce et al., 1997*; *Briones-Fourzán et al., 2007*).

Based on the general positive or neutral effects of casitas on lobsters and their habitats, the use of scaled-down casitas has further been suggested to enhance juvenile lobsters in marine protected areas (*Sosa-Cordero et al., 1998*; *Briones-Fourzán, Lozano-Álvarez & Eggleston, 2000*; *Briones-Fourzán et al., 2007*; *Briones-Fourzán & Lozano-Álvarez, 2013*). But in 1999-2001, a previously unknown disease emerged in populations of juvenile Caribbean spiny lobsters. The disease is caused by Panulirus argus virus 1 (PaV1) (*Shields & Behringer, 2004*), a member of the newly established family Mininucleoviridae (*Subramaniam et al., 2020*). PaV1 can be lethal for juveniles of *P. argus* ($\leq$ 50 mm carapace length, CL), for which this virus shows predilection (*Shields & Behringer, 2004*; *Li et al., 2008*).

PaV1 can be transmitted by contact and through water, at least over distances of 1–2 m, raising concerns about the potential increase in transmission with the use of casitas (*Butler, Behringer & Shields, 2008*; *Behringer & Butler, 2010*; *Behringer et al., 2012*). However, *P. argus* express behavioral immunity, *i.e.*, a tendency to avoid odors emanating from infected conspecifics (*Behringer, Butler & Shields, 2006*; *Candia-Zulbarán et al., 2015*), which may help reduce the transmission of the disease (*Anderson & Behringer, 2013*; *Butler et al., 2015*). Yet, *Lozano-Álvarez et al. (2008)* observed high levels of cohabitation between healthy and diseased lobsters in experimental scaled-down casitas deployed over shelter-limited seagrass habitats. These authors hypothesized that, on these habitats, lobsters make a trade-off between avoiding disease and avoiding predation risk, and that the large shelter area provided by casitas may reduce physical contact among healthy and diseased lobsters. Upon testing these hypotheses, *Lozano-Álvarez et al. (2018)* found that both the availability of alternate shelter and immediacy of predation risk modulate the expression of behavioral immunity in *P. argus*. They also found that healthy lobsters tended to be segregated from co-occurring diseased lobsters in casitas, although this distancing decreased with increasing number of lobsters in a casita. In a different study, distribution parameters of lobsters in large commercial casitas were generally not affected by the presence of diseased conspecifics; rather, investment in disease avoidance by lobsters appeared to be partially modulated by local habitat features (*Briones-Fourzán et al., 2012*).

In addition to behavioral immunity reducing transmission of PaV1, *Butler et al. (2015)* also contemplated the possibility of predatory culling of diseased lobsters, which has been shown to reduce the spread of pathogens in some systems, but not in others (review in

*Lopez & Duffy, 2021*). Culling diseased individuals during surveillance or fishing operations has also been suggested as a means to reduce disease transmission if the culled individuals are disposed of at land (*Behringer et al., 2012*). However, whether culling is an efficient way to manage marine diseases is still a matter of debate (*Groner et al., 2016*; *Behringer et al., 2020*; *Glidden et al., 2022*).

The studies supporting the suggestion to use scaled-down casitas to enhance juvenile lobsters in shelter poor habitats were conducted before the full establishment of PaV1, but as juvenile lobsters are more susceptible to PaV1 than adults, further investigation is required on the potential effects of casitas on the dynamics of the PaV1 disease. We addressed this issue *via* a field experiment consisting of two stages. In the first stage, we examined the relationship between the expected increase in density of juvenile lobsters with scaled-down casitas and the prevalence of PaV1 disease, and compared the patterns of shelter occupancy by lobsters as related to disease between casitas and natural crevices. We did not expect disease prevalence to increase with the use of casitas despite increasing lobster density given the complex but flexible behavioral responses of *P. argus* under different ecological contexts. In the second stage, we tested whether the systematic culling of diseased lobsters from casita sites altered the probability of disease and whether such changes were consistent among sites.

## MATERIALS & METHODS

### Study area

The study was conducted in the reef lagoon of the Puerto Morelos Reef National Park, a marine protected area located on the northern part of the Mexican Caribbean coast. The reef lagoon (centered at 20°52′07″N, 86°51′40″W) extends from the shore to the coral reef tract, which lies at ∼500 m to 2,000 m from the shore. Maximum depth within the reef lagoon is 5 m (Fig. 1). No lobster fishing is allowed within the reef lagoon.

The Puerto Morelos reef lagoon has been extensively studied since the early 1990s (*e.g.*, *Rodríguez-Martínez et al., 2010*; *Caballero-Aragón et al., 2022*). Based on its vegetation, the lagoon is divided into a narrow coastal fringe (50–100 m in width), a broad mid-lagoon zone, and a back-reef lagoon zone. The present study took place in the mid-lagoon, which encompasses the greatest part of the lagoon, and the back-reef lagoon. In the mid-lagoon, the sandy sediments tend to be deeper and the seagrass biomass and height are generally greater, but with substantial temporal and spatial variation. In the back-reef lagoon, seagrass meadows have generally less biomass, shorter leaves, and a less dense canopy because the sediment layer is thinner and hard substrate is more abundant (*van Tussenbroek, 2011*; *Zarco-Perelló & Enríquez, 2019*). Biomass of drift algae tends to be greater on the mid-lagoon than on the back-reef lagoon zone (*Van Tussenbroek, 2011*; *Lozano-Álvarez, Meiners & Briones-Fourzán, 2009*). Although the reef lagoon constitutes a nursery area for juveniles of *P. argus* (≤50 mm CL), shelter is a limiting factor for the larger juveniles (*Briones-Fourzán et al., 2007*) because crevice-type shelter is scarce and over-dispersed in the reef lagoon (*Briones-Fourzán & Lozano-Álvarez, 2001*).

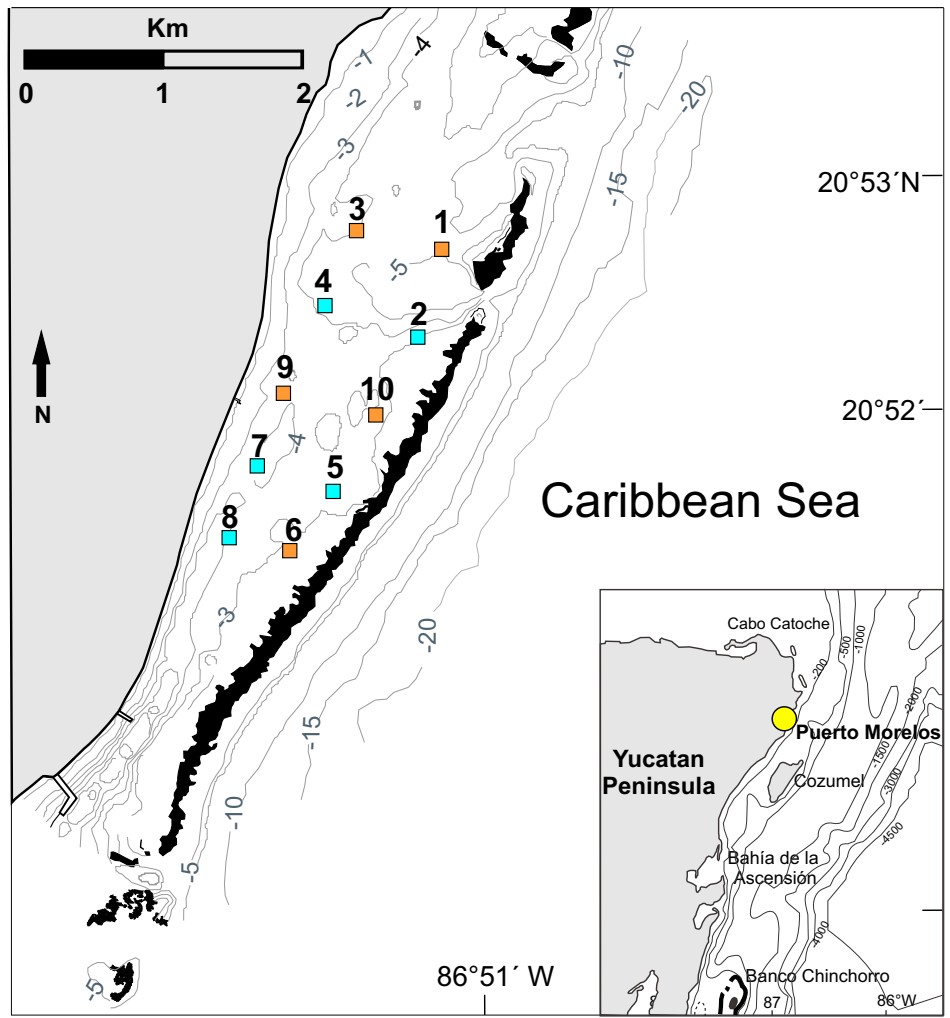

**Figure 1 Study area.** Location of the 10 experimental sites on the Puerto Morelos reef lagoon. Each site measured 1 ha (100 m × 100 m). Control sites (orange squares: sites 1, 3, 6, 9 and 10) had no casitas. Casita sites (blue squares: sites 2, 4, 5, 7 and 8) had 10 casitas each. Black areas denote coral reefs. Inset denotes location of Puerto Morelos (yellow dot). Isobaths are in meters. Figure modified from *Davies et al. (2020)*.

## Experimental design

Permits to conduct this study were issued by Comisión Nacional de Acuacultura y Pesca (DGOPA.12019.031108.3134, DGOPA-06695.190612.1737, and PPF/DGOPA-259/14). The experimental design followed that of *Briones-Fourzán & Lozano-Álvarez (2001)* and *Briones-Fourzán et al. (2007)*. Briefly, 10 experimental sites were delimited in the reef lagoon (Fig. 1). Each site measured 100 m × 100 m (= 1 ha), an area that exceeds the daily home range of juveniles of *P. argus* (<1 to ~20 m) (*Butler, Steneck & Herrnkind, 2006*; *Lozano-Álvarez, Meiners & Briones-Fourzán, 2009*). To ensure the independence of data, all sites were separated from each other and from the reef tract by distances of 200 m to 600 m, which exceed the movement range of juveniles ≤70 mm CL (*Briones-Fourzán et*

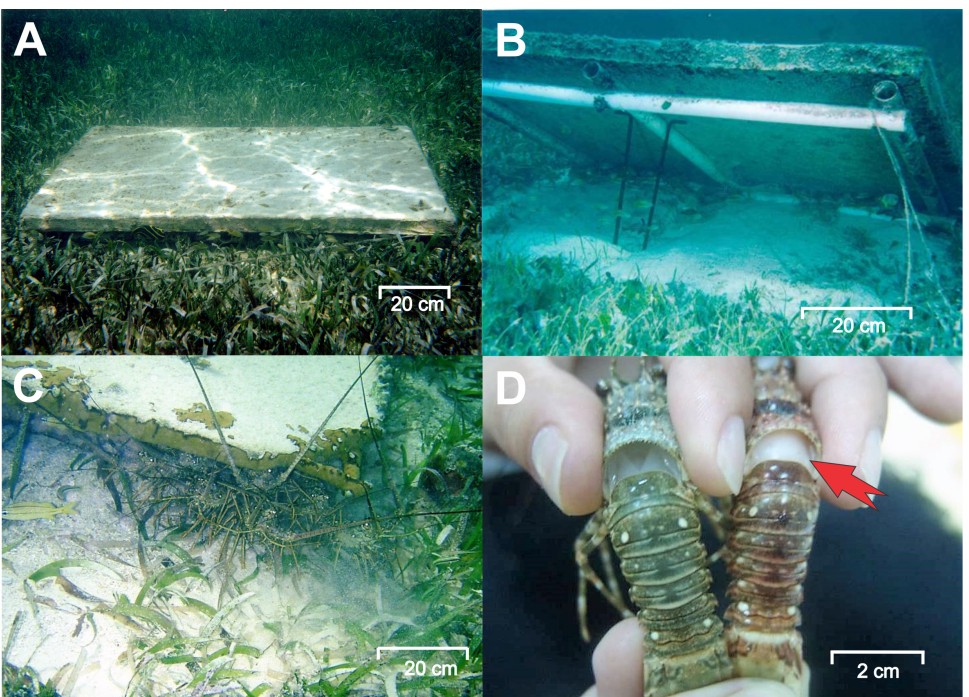

**Figure 2 Experimental casitas and clinical signs of PaV1.** (A) An experimental casita (1.1 m² in area, 4 cm entrance height, 8 cm inner height) deployed on the bottom. (B) A casita lifted to show its frame. (C) Lobsters sheltering beneath a casita. (D) A healthy lobster (left) and a diseased lobster (right) exhibiting clinical signs of PaV1: milky hemolymph (red arrow) and a reddish discoloration of the exoskeleton. Photo credits: Fernando Negrete-Soto.

*al., 2007*). In August 2009, we deployed 10 scaled-down casitas per site on five randomly chosen sites (sites 2, 4, 5, 7, and 8 in Fig. 1, hereafter "casita sites"), whereas the other five sites remained without casitas (sites 1, 3, 6, 9, and 10 in Fig. 1, hereafter "control sites"). Casita dimensions were 1.1 m² in area ×4 cm in entrance height and eight cm in inner height (Figs. 2A–2C). On each site, the 10 casitas were randomly deployed, but leaving a distance of at least 20 m between adjacent casitas. To facilitate working within each site, wood stakes were installed every 10 m throughout the site area.

## Experimental stage A

Experimental stage A was designed to examine the potential effect of casita deployment on prevalence of the PaV1 disease as well as the occupancy patterns of shelters (both casitas and natural crevices) by lobsters as related to disease. *Briones-Fourzán et al. (2007)* found that density and mean size of lobsters significantly increased on casita sites relative to control sites, but their study ended in 2002, when PaV1 was just emerging. Because the present study was conducted after several years of increasing prevalence of PaV1 (*Lozano-Álvarez et al., 2008*), in addition to clinical prevalence of PaV1 we also compared mean density and size of lobsters between casita sites and control sites.

*Briones-Fourzán et al. (2007)* estimated a persistence of 80.9 ± 17.8 d (mean ± SD) for juvenile lobsters on casita sites and of 40.7 ± 10.3 d on control sites. Therefore,

between November 2009 and April 2012, we conducted 10 samplings every 3–4 months to minimize the possibility of serial correlation of data. Using scuba, samplings consisted of surveying the entire area of each site for lobsters, including beneath casitas and in all pre-existing shelters, such as hard coral heads, sponges, soft corals, outcrops, solution holes and all kinds of crevices. The largest external diameter of a natural shelter is considered a good approximation to the shelter area it provides (*Childress & Herrnkind, 1997*; *Briones-Fourzán & Lozano-Álvarez, 2001*); therefore, pre-existing shelters were categorized into small (<25 cm across the largest external diameter), medium (25–50 cm), or large (>50 cm). Lobsters were extracted from their crevice or casita with hand nets and visually examined for clinical (macroscopic) signs of PaV1 infection (milky hemolymph, visible through the translucid membrane between the cephalothorax and abdomen, and a reddish discoloration of the clear marks over the exoskeleton, Fig. 2D) (*Shields & Behringer, 2004*; *Huchin-Mian et al., 2008*). Lobsters with these signs are hereafter referred to as "diseased" (*Montgomery-Fullerton et al., 2007*; *Li et al., 2008*; *Candia-Zulbarán et al., 2012*; *Huchin-Mian et al., 2013*). Specificity and sensitivity of the macroscopic determination of PaV1 estimated against endpoint PCR were 1.0 and 0.5, respectively, both in the Puerto Morelos reef lagoon, where mostly juvenile lobsters are found (*Candia-Zulbarán, Briones-Fourzán & Lozano-Álvarez, 2019*), and in Bahía de la Ascensión, where lobsters span a greater size range (*Huchin-Mian et al., 2013*). Therefore, in these areas, for every visibly diseased lobster there is another subclinically infected lobster (*i.e.,* lobsters carrying the virus but without having developed the disease). However, for the sake of simplicity all lobsters with no clinical signs of PaV1 are hereafter referred to as "healthy". The carapace length (CL, mm) of lobsters was measured from the inter-orbital notch to the rear end of the carapace with Vernier calipers. Examination and measurement of lobsters was conducted *in situ* (underwater) to avoid exposure to air and to reduce handling stress. After data collection, all lobsters were carefully returned to their previous shelter or casita. For each site and sampling time, disease prevalence was estimated as the number of diseased lobsters over the total number of lobsters ×100.

## Experimental stage B

Experimental stage B was intended to examine whether culling all diseased lobsters found on each sampling date altered overall prevalence levels relative to stage A, and whether such changes were consistent among sites. For these purposes, from September 2012 to January 2015 we conducted eight additional samplings every 3–4 months. On each of these samplings, all lobsters with clinical signs of PaV1 were culled (*i.e.,* removed and taken to land). Stage B was conducted exclusively on casita sites. Control sites were not considered in this stage because of their extreme paucity of lobsters (see Results).

## Data analyses
### Experimental stage A

*Lobster density and mean size*—The data on lobster density (previously transformed to Log (N + 1) to increase homogeneity of variances) and lobster size were separately subjected to repeated measures analysis of variance using a General Lineal Model (GLM) approach. The
main (fixed) factor was site type (with two levels, casita sites and control sites), whereas time (10 samplings between November 2009 and April 2012) was the repeated measure.

*Relationship between lobster density and disease prevalence.* Initially, we had planned to use a logistic regression analysis in which the binary response variable would be the absence/presence of clinical signs of PaV1, to examine the effect of site type and sampling time (categorical factors) on the probability of finding diseased lobsters (*Quinn & Keough, 2002*). Unfortunately, the complete lack of lobsters or the absence of diseased lobsters on one or more control sites on several sampling times (Table S1) precluded the use of this analysis, as the model required data on lobsters with and without clinical signs of PaV1 in all levels of both factors. Instead, we used correlation analyses to examine whether disease prevalence tended to increase with lobster density (lobsters ha$^{-1}$). For this analysis, we considered the data from all casita sites and control sites on which the number of lobsters on a given sampling time was $\geq 5$ (*Putt et al., 1988*). Separate analyses were performed for casita sites, for control sites, and for all sites together. We further compared disease prevalence between site types with a Mann–Whitney test.

*Occupancy of shelters as related to diseased individuals* —Preliminary analyses revealed that very few small and medium pre-existing shelters were occupied by lobsters, both on casita sites (0.6% and 7.0%, respectively) and on control sites (3.9% and 16%, respectively), and that in most cases these shelters, when occupied, harbored a solitary lobster. In contrast, 33.0% of all large pre-existing shelters on casita sites, and 37.2% on control sites, were occupied by one or multiple lobsters. Therefore, only large pre-existing shelters (hereafter "crevices") were considered for comparison with lobster distribution in casitas.

To determine whether the pattern of distribution of lobsters among crevices and casitas varied as a function of the presence/absence of diseased lobsters in the shelter, we followed the model selection approach used by *Briones-Fourzán et al. (2012)* for commercial casitas, based on the negative binomial distribution (NBD) of the number of lobsters per shelter. The following procedure was separately applied to casitas and crevices on casita sites, and to crevices on control sites, but is only explained for casitas.

The NBD is defined by two parameters: *m*, which is the average number of lobsters per casita (or crevice) and *k*, which is a dispersion parameter. As *k* tends to infinity, the distribution approaches a random distribution and the data can be modeled as a Poisson process, whereas as *k* tends to zero, the distribution becomes more clumped (*White & Bennets, 1996*). Model selection uses a likelihood ratio testing framework to identify the best model out of a set of competing models to explain selected parameters for a given set of samples (*Burnham & Anderson, 2002*). We separated the total number of casitas examined over experimental stage A into two samples: one consisting of casitas containing exclusively healthy lobsters, and one consisting of casitas containing healthy + diseased lobsters. We then used a set of four candidate models to compare parameters of the NBD of lobsters over our sampling times (*White & Eberhardt, 1980*). The general (most parameterized) model, $\{k_v, m_v\}$, predicts that all samples *v* differ in *m* and *k*. The other three models, which have fewer parameters, are $\{k, m_v\}$: samples have a common *k* but different *m*; $\{k_v, m\}$: samples have a different *k* but a common *m*, and $\{k, m\}$: all samples have a common *k* and a common *m* (the reduced model) (*White & Eberhardt, 1980*; *White & Bennets, 1996*).

The "best" model would be that with the lowest Akaike Information Criterion corrected for sample size (AICc, see *Burnham & Anderson, 2002*). However, delta AICc ($\Delta_i$) and the Akaike weight ($w_i$) provide better measures of the strength of evidence for each model. $\Delta_i$ is the AICc of a given model minus the AICc of the best model (whose $\Delta i$ is set to zero), whereas $w_i$ represents the ratio of the $\Delta_i$ of a given model relative to the whole set of models (the $w_i$ from all models sum to 1) and thus represents the "probability" of each model given the data. Akaike weights also provide a basis for model averaging, a procedure that allows the entire set of models to be used to compute a weighted average for each parameter and its corresponding unconditional variance (*Hobbs & Hillborn, 2006*).

The analysis was done with the software EcoMeth 6.1 (*Kenney & Krebs, 2002*), which includes a modification of the computer program originally developed by *White & Eberhardt (1980)* and *White & Bennets (1996)*. The output from this software provides, for each model, the goodness of fit to the NBD, maximum likelihood and AIC values, and the corresponding estimates of $m$ and $k$ with their respective variances. These estimates allow the computation of AICc, $\Delta_i$, and $w_i$, as well as the model averaging procedure (see *Burnham & Anderson, 2002*; *Hobbs & Hillborn, 2006*).

### Experimental stage B

Data from experimental stage B were used to analyze the effects of culling and site on the probability of finding disease. When the response variable is binary (*e.g.*, presence/absence of clinical signs of PaV1), the appropriate test is a logistic regression analysis, which is a type of generalized linear model (*Quinn & Keough, 2002*). The predictors (categorical factors) in the model were experimental stage and casita site. Site was included in the model to account for the potential effects of local habitat characteristics. Experimental stage had two levels (stage A, with no culling, and stage B, with culling), with stage A as the baseline (reference) level. Site had 5 levels (corresponding to the five casita sites), with site 8 as the reference level. This analysis revealed whether culling all diseased lobsters throughout experimental stage B altered disease probability relative to stage A and whether these changes were consistent among casita sites.

## RESULTS

### Experimental stage A

*Lobster density and size*—Lobster density was significantly affected by site type and sampling time, but not by their interaction (Table 1). On average, lobster density was about eight times as high on casita sites (overall mean ± SD: 72.4 ± 46.3 lobsters ha$^{-1}$) as on control sites (8.7 ± 1.7 lobsters ha$^{-1}$), but with substantial temporal variation on both site types (Fig. 3A). Similarly, lobster size was significantly affected by site type and sampling time, but not by their interaction (Table 1). The mean size of lobsters was generally higher on casita sites (overall mean ± SD: 30.8 ± 3.3 mm CL) than on control sites (23.6 ± 2.8 mm CL), but varied over time on both site types (Fig. 3B).

*Relationship between lobster density and disease prevalence*—Throughout experimental stage A, there were 50 data on lobster density and disease prevalence for casita sites (5 sites ×10 sampling times), but only 34 data for control sites, because 16 data corresponded to

**Table 1 Effect of casitas and time on lobster density and size.** Results of General Lineal Models testing for effects of site type (two levels: casita sites and control sites) and time (10 sampling dates, repeated measure) on density of lobsters (Log (N +1) lobsters per ha) and lobster size (carapace length, mm).

| Effect | df | Lobster density | | | Lobster size | | |
|---|---|---|---|---|---|---|---|
| | | MS | F | *p* | MS | F | *p* |
| Intercept | 1 | 921.177 | 320.952 | <0.001 | 1061.678 | 5590.229 | <0.001 |
| Site type | 1 | 112.931 | 39.347 | <0.001 | 1.774 | 9.341 | 0.016 |
| Error | 8 | 2.870 | | | 0.190 | | |
| Time | 9 | 1.420 | 6.616 | <0.001 | 0.081 | 3.418 | 0.002 |
| Time × Site type | 9 | 0.346 | 1.614 | 0.127 | 0.041 | 1.726 | 0.099 |
| Error | 72 | 0.215 | | | 0.024 | | |

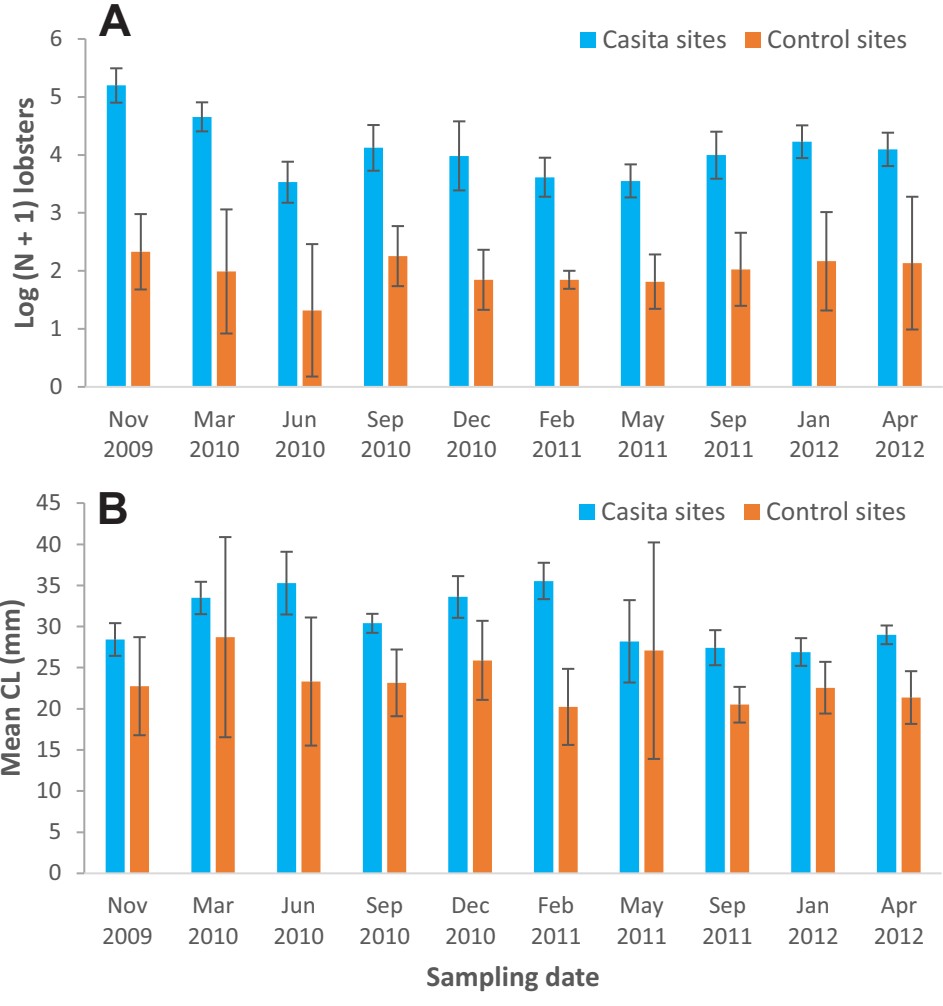

**Figure 3 Mean lobster density and lobster size.** (A) Lobster density (Log (N + 1)) and (B) lobster mean size (carapace length, mm) on casita sites (blue columns) and control sites (orange columns) throughout experimental stage A. Error bars are 95% confidence intervals.

sites with 0–4 lobsters on a given sampling time and hence could not be included (Table S1). Throughout experimental stage A, overall disease prevalence was 15.9% ± 7.4% on casita sites (overall mean ± SD, $N = 50$) and 14.4% ± 4.0% on control sites ($N = 34$). These values were not significantly different (Mann–Whitney test, $U = 694.5$, $p = 0.157$). The correlation between log-transformed lobster density and disease prevalence was not significant on casita sites ($r = -0.217$, $N = 50$, $p = 0.130$) (Fig. 4A), on control sites ($r = 0.009$, $N = 34$, $p = 0.962$) (Fig. 4B), or on both types of sites together ($r = 0.005$, $N = 89$, $p = 0.962$) (Fig. 4C).

*Occupancy of shelters as related to disease*— Throughout experimental stage A, we examined, 500 casitas (50 casitas ×10 sampling times) and 215 crevices on casita sites, and 495 crevices on control sites. Few casitas and crevices contained only diseased lobsters. However, most casitas (45.4%) harbored heathy + diseased lobsters, followed by casitas with only healthy lobsters (34.8%), whereas 11.2% casitas had no lobsters (Table 2). In contrast, most crevices in both casita sites and control sites harbored zero lobsters (67.5% and 62.8%, respectively), followed by crevices occupied exclusively by healthy lobsters (22.3% and 28.7%, respectively), whereas crevices harboring both healthy + diseased lobsters were scarce (5.1% and 3.6%, respectively) (Table 2). On casita sites, the maximum number of occupants was 23 for casitas with healthy lobsters, 107 for casitas with healthy + diseased lobsters, 10 for crevices with healthy lobsters, and five for crevices with healthy and diseased lobsters. On control sites, crevices with healthy lobsters and crevices with healthy + diseased lobsters had a maximum of eight and five lobsters, respectively.

In all cases, the NBD fitted well the distribution of lobsters. On casita sites, $\{k, m_v\}$ and $\{k_v, m_v\}$ were the best models for casitas as well as for crevices (Table 3). Both models estimated a different $m$ for each sample but the former estimated a common $k$ for all samples, whereas the latter estimated a different $k$ for each one. Therefore, we proceeded with model averaging. Casitas with healthy + diseased lobsters had a higher $m$ (7.08 ± 0.49 lobsters per casita, mean ± SE) than casitas with healthy lobsters (4.07 ± 0.34 lobsters per casita) (Fig. 5A), whereas $k$ did not differ significantly between both casita samples (0.75 ± 0.08 and 0.72 ± 0.06, respectively) (Fig. 5B). Throughout casita sites, crevices with healthy lobsters had a higher $m$ (0.62 ± 0.13 lobsters per crevice) than crevices with healthy + diseased lobsters (0.31 ± 0.10 lobsters per crevice), whereas $k$ did not differ significantly between both crevice samples (0.19 ± 0.04 and 0.15 ± 0.05, respectively). Therefore, on casita sites, both $m$ and $k$ were higher in casitas than in crevices; that is, casitas in general harbored more lobsters, but the distribution of lobsters was more clumped in crevices.

On control sites, by contrast, model $\{k_v, m_v\}$ was by far the best fit to the data on lobster distribution, with a $w_i$ of 0.999 (Table 4), with crevices with healthy lobsters having higher values of $m$ (0.61 ± 0.06 lobsters per crevice) and $k$ (0.43 ± 0.07) than crevices with healthy + diseased lobsters ($m = 0.16 ± 0.05$ lobsters per crevice; $k = 0.11 ± 0.03$) (Figs. 5A and 5B).

## Experimental stage B

On the five casita sites, we examined 5,714 lobsters in total, 3,619 during experimental stage A and 2,095 during stage B. Of the total lobsters observed in stage B, all diseased

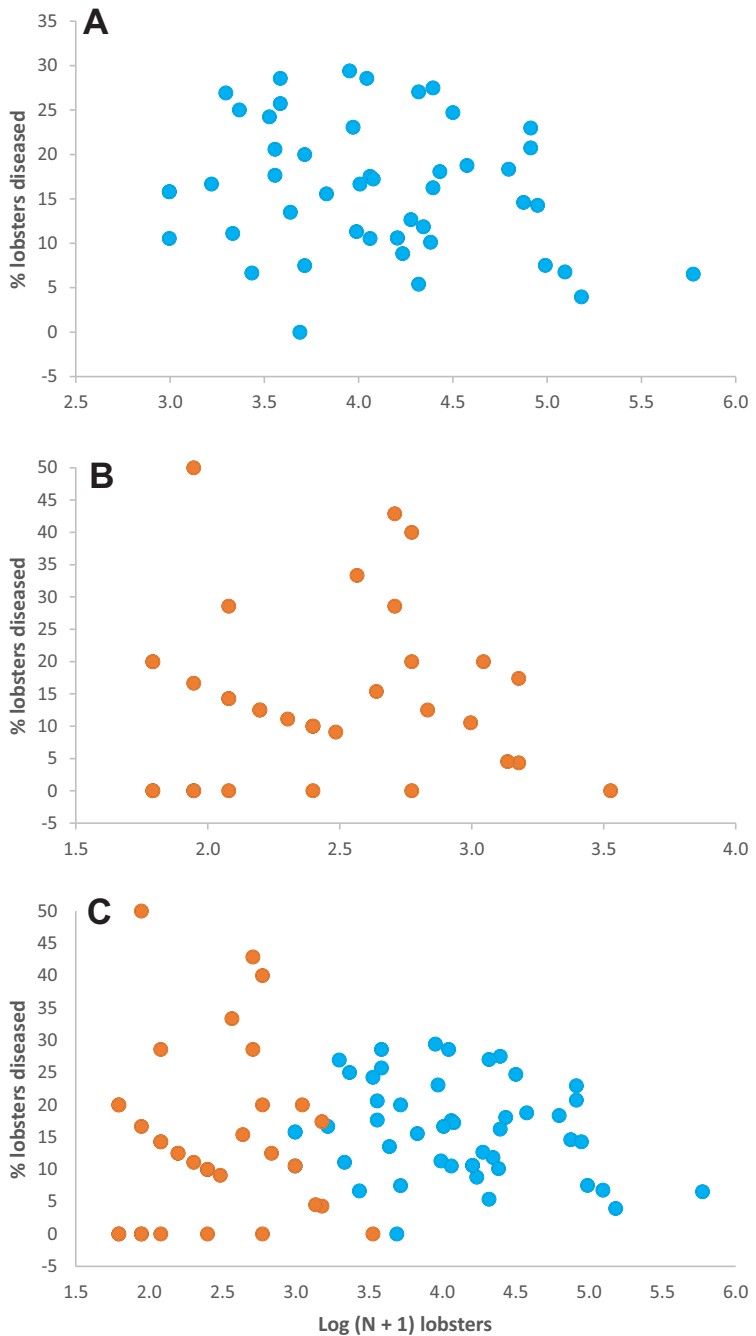

**Figure 4 Lobster density *versus* disease prevalence.** Relationship between lobster density (Log (N + 1) lobsters ha$^{-1}$) and prevalence of PaV1 disease (percentage of diseased lobsters) in (A) casita sites (blue dots), (B) control sites (orange dots), and (C) all sites throughout experimental stage A.

lobsters (442 in total) were culled (Table S1). The logistic regression analysis showed that the probability of finding diseased lobsters was higher in stage B (odds ratio: 1.2) relative to stage A and varied significantly with site (Table 4). Compared to site 8 (the reference

**Table 2** **Occupancy of casitas and crevices.** Summary of data from casitas and crevices distributed on casita sites, and from crevices distributed on control sites, occupied by healthy lobsters, diseased lobsters, healthy + diseased lobsters, and no lobsters.

| Lobster condition | Casita sites | | | | Control sites | |
|---|---|---|---|---|---|---|
| | Casitas | % | Crevices | % | Crevices | % |
| Healthy | 172 | 34.4 | 48 | 22.3 | 142 | 28.7 |
| Diseased | 45 | 9.0 | 12 | 5.6 | 24 | 4.8 |
| Healthy + diseased | 227 | 45.4 | 11 | 5.1 | 18 | 3.6 |
| No lobsters (empty) | 56 | 11.2 | 144 | 67.0 | 311 | 62.8 |
| Total | 500 | 100 | 215 | 100 | 495 | 100 |

**Table 3** **Model selection for occupancy of casitas and crevices.** Results of model selection contrasting four models (based on parameters of the negative binomial distribution) (A) for casita sites, separately considering a set of two casita samples (casitas occupied exclusively by healthy lobsters and casitas co-occupied by healthy and diseased lobsters) and a set of two crevice samples (crevices occupied exclusively by healthy lobsters and crevices co-occupied by healthy and diseased lobsters), and (B) for control sites, considering a set of two crevice samples (crevices occupied exclusively by healthy lobsters and crevices co-occupied by healthy and diseased lobsters).

| | Model | No. of parameters | Maximum likelihood | $AIC_c$ | $\Delta AIC_c$ | $w_i$ |
|---|---|---|---|---|---|---|
| **(A) Casita sites** | | | | | | |
| | $\{k, m_v\}$ | 3 | −1528.960 | 3063.964 | 0 | 0.6588 |
| Casitas | $\{k_v, m_v\}$ | 4 | −1528.603 | 3065.280 | 1.316 | 0.3412 |
| | $\{k, m\}$ | 2 | −1541.292 | 3086.606 | 22.642 | 0 |
| | $\{k_v, m\}$ | 3 | −1541.233 | 3088.510 | 24.546 | 0 |
| | $\{k, m_v\}$ | 3 | −291.230 | 588.528 | 0 | 0.5093 |
| Crevices | $\{k_v, m_v\}$ | 4 | −290.505 | 589.123 | 0.595 | 0.3782 |
| | $\{k_v, m\}$ | 3 | −293.333 | 592.734 | 4.206 | 0.0622 |
| | $\{k, m\}$ | 2 | −294.561 | 593.156 | 4.628 | 0.0503 |
| **(B) Control sites** | | | | | | |
| | $\{k_v, m_v\}$ | 4 | −660.027 | 1328.104 | 0 | 0.9988 |
| Crevices | $\{k_v, m\}$ | 3 | −668.051 | 1342.132 | 14.028 | 0.0009 |
| | $\{k, m_v\}$ | 3 | −669.150 | 1344.330 | 16.226 | 0.0003 |
| | $\{k, m\}$ | 2 | −682.953 | 1369.921 | 41.817 | 0 |

**Notes.**

AICc, Akaike information criterion adjusted for small sample size; ΔAICc, difference between each AICc and the smallest AICc; wi, Akaike weight.

site), the probability of finding diseased lobsters was overall higher on site 4 (odds ratio: 1.31), and lower on site 7 (odds ratio: 0.71), but did not differ significantly on sites 2 and 5 (odds ratio: 1.1 and 0.9, respectively) (Table 4). Disease prevalence on casita sites varied with sampling time in both stages but tended to be higher throughout stage B (Fig. 6A). However, relative to stage A, average disease prevalence during stage B was significantly higher only on casita sites 2 and 4 (Fig. 6B).

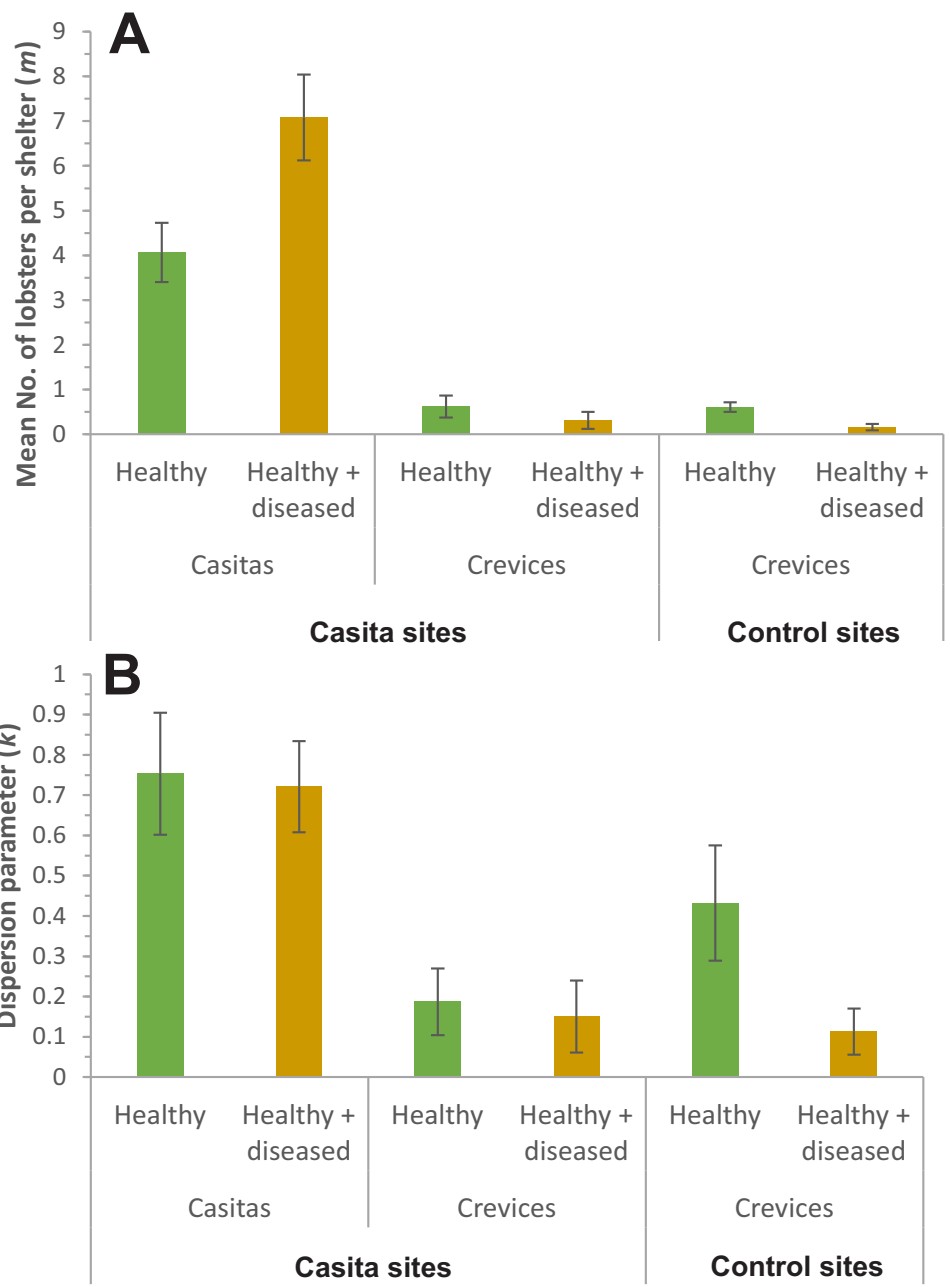

**Figure 5** **Parameters of the negative binomial distribution.** Comparison of (A) $m$ (mean number of lobsters per casita or crevice) and (B) $k$ (dispersion parameter) for casitas and crevices on casita sites, and for crevices on control sites, occupied exclusively by healthy lobsters (green columns) or co-occupied by healthy and diseased lobsters (brown columns). Error bars are 95% confidence intervals.

## DISCUSSION

We examined the potential effects of scaled-down casitas for juvenile lobsters on the dynamics of the PaV1 disease. Due to the larger aggregations of lobsters with increasing

**Table 4  Logistic regression analysis.** Estimates for logistic regression analysis testing the effects of experimental stage (two levels: Stage A (without culling), Stage B (with culling of diseased lobsters); Stage A is the reference level) and casita site (five levels: Sites 2, 4, 5, 7 and 8; site 8 is the reference level) on the probability of finding diseased lobsters.

| | Effect level | Estimate | Standard error | Wald statistic | $p$ | Odds ratio (IC 95%) |
|---|---|---|---|---|---|---|
| Intercept | | −1.570 | 0.039 | 1639.106 | <0.001 | |
| Stage | Stage B | 0.179 | 0.039 | 21.337 | <0.001 | 1.20 (1.11–1.29) |
| Site | 2 | 0.097 | 0.071 | 1.905 | 0.168 | 1.10 (0.96–1.27) |
| Site | 4 | 0.268 | 0.067 | 16.078 | <0.001 | 1.31 (1.15–1.49) |
| Site | 5 | −0.130 | 0.077 | 2.852 | 0.091 | 0.88 (0.75–1.02) |
| Site | 7 | −0.341 | 0.096 | 12.578 | <0.001 | 0.71 (0.59–0.86) |

lobster density, disease prevalence could be expected to increase over time on casita sites, where there were on average eight times as many lobsters as on control sites, but we found no apparent correlation between lobster density and disease prevalence. These experimental results are akin to results from different simulated scenarios of host spatial structure and avoidance of diseased lobsters by healthy conspecifics, which showed no increase in transmission or persistence of PaV1 with increasing density of lobsters (*Dolan, Butler & Shields, 2014*).

Under experimental conditions, healthy lobsters avoid shelters harboring diseased lobsters (*Behringer, Butler & Shields, 2006*; *Candia-Zulbarán et al., 2015*), but whether and to what extent this occurs in natural conditions likely depends on the ecological context (*Butler et al., 2015*; *Lozano-Álvarez et al., 2018*). Predation risk for juveniles of *P. argus* is ever present (*Smith & Herrnkind, 1992*; *Butler, Steneck & Herrnkind, 2006*), especially in seagrass meadows where crevice shelters are scarce (*Briones-Fourzán & Lozano-Álvarez, 2001*). In the present study, 11% casitas, but ∼65% crevices, did not harbor any lobsters, reflecting the smaller effective refuge area provided by crevices (even the larger ones) relative to casitas. In these circumstances, lobsters may avoid shelters with limited space already occupied by diseased conspecifics, although this would increase their predation risk (*Anderson & Behringer, 2013*). But if casitas are deployed in those habitats, their large refuge area may allow segregation of healthy and diseased lobsters (*Lozano-Álvarez et al., 2018*). *Gutzler, Butler & Behringer (2015)* cautioned that a concentration of small lobsters in casitas may increase the abundance of large piscine predators. However, lobsters and predators typically cohabit in casitas on account of their large refuge area (*Lozano-Álvarez & Spanier, 1997*; *Sosa-Cordero et al., 1998*; *Lozano-Álvarez et al., 2010*; *Briones-Fourzán et al., 2012*; *Ross, Butler & Matthews, 2022*). Moreover, competition for scarce shelter between lobsters and other taxa, including predators, can be reduced if shelter availability increases, *e.g.*, with casitas (*Lozano-Álvarez et al., 2010*; *Briones-Fourzán et al., 2012*).

The analyses of lobster distribution among casitas and crevices yielded interesting results. The mean number of lobsters ($m$) was higher in casitas than in crevices, but the distribution of lobsters was more clumped ($k$ was closer to zero) in crevices than in casitas. However, $m$ was much higher in casitas co-occupied by healthy + diseased lobsters than in casitas harboring only healthy lobsters, even though $k$ was similar in both

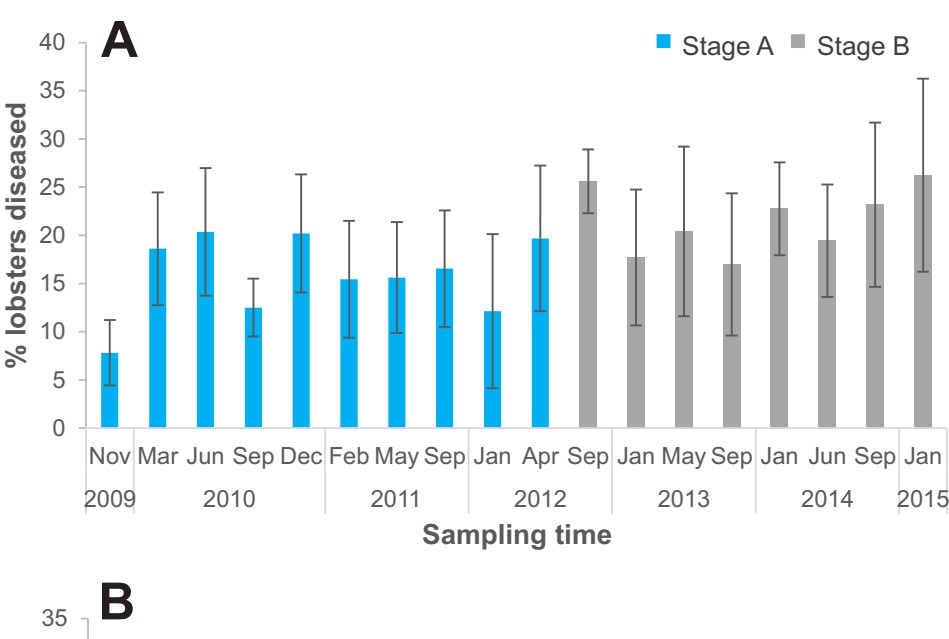

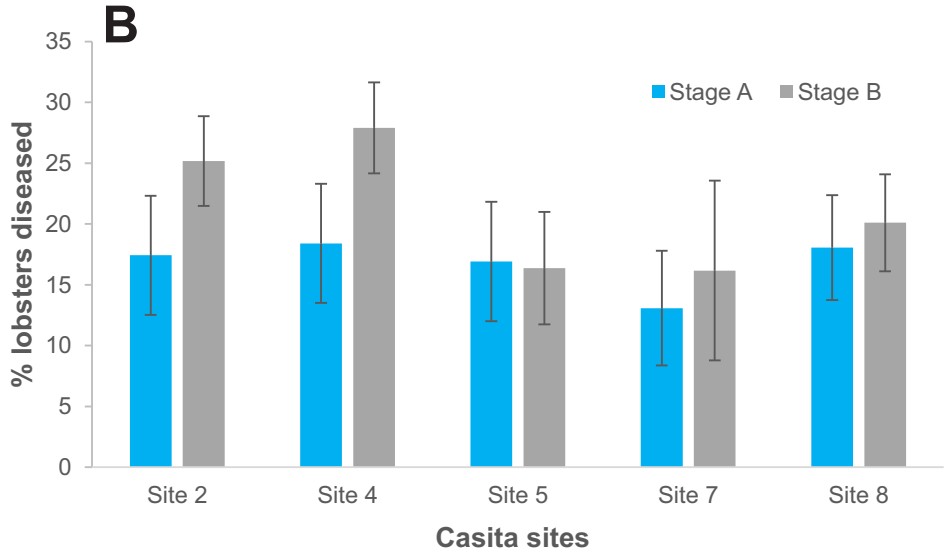

**Figure 6** **Disease prevalence on casita sites by experimental stage.** (A) Prevalence of the PaV1 disease (percentage of diseased lobsters) on casita sites by sampling time throughout experimental stages A (blue columns) and B (gray columns). (B) Average percentage of diseased lobsters on individual casita sites during experimental stages A (blue columns) and B (gray columns) Error bars are 95% confidence intervals.

casita samples. In conjunction, these findings confirm the paucity of pre-existing shelters throughout the Puerto Morelos reef lagoon and that casitas do allow for the cohabitation of healthy and diseased lobsters. Values of $m$ and $k$ were also generally not affected by the presence of diseased lobsters in large commercial casitas throughout Bahía de la Ascensión (*Briones-Fourzán et al., 2012*).

In contrast with casitas, cohabitation of healthy and diseased lobsters was less common in crevices, and $m$ was smaller in crevices harboring healthy + diseased lobsters than in those harboring only healthy lobsters. These results confirm the tendency of healthy

lobsters to avoid sharing natural crevice-type shelters with diseased conspecifics (*Butler et al., 2015*), likely because they provide a smaller space than casitas. Similarly, in the lower Florida Keys (USA), significantly more lobsters were found in casitas than at either coral heads or low relief hardbottom (*Ross, Butler & Matthews, 2022*).

Although healthy lobsters tend to avoid diseased lobsters, the latter maintain their gregarious behavior (*Behringer & Butler, 2010*). *Lozano-Álvarez et al. (2018)* found that the tendency of healthy lobsters to stay away from diseased lobsters beneath casitas decreased as the number of lobsters per casita increased, suggesting that the large refuge area provided by casitas allows the segregation between healthy and diseased lobsters only to a certain point. This segregation will depend both on the number of lobsters occupying the casita and on the gregarious behavior expressed by the diseased lobsters (*Lozano-Álvarez et al., 2018*), which is why we tested whether culling diseased individuals may favor aggregation of lobsters and reduce prevalence levels.

Contrary to expectation, the overall level of prevalence on casita sites increased during experimental stage B, when culling was performed, relative to stage A. However, we only culled all overtly diseased lobsters, which are the most infective (*Li et al., 2008*), every 3–4 months, a period during which some subclinically infected lobsters likely developed the disease and other healthy lobsters could have become infected. Although subclinically infected lobsters are as abundant as clinically infected lobsters (*Candia-Zulbarán, Briones-Fourzán & Lozano-Álvarez, 2019*), they are not equally avoided by healthy conspecifics (*Candia-Zulbarán et al., 2015*). Therefore, culling more often than we did could potentially yield different results (*Groner et al., 2016*). On the other hand, selective predation on infected prey does not always reduce infection prevalence (*Lopez & Duffy, 2021*), and some models have shown that when the most heavily infected individuals in a population are culled, disease prevalence may increase due to persistence of less virulent strains of the parasite which are able to establish in sparser populations (*Bolzoni & De Leo, 2013*; *Behringer et al., 2020*). Indeed, many authors do not consider culling an effective way to manage marine infectious diseases because of the dearth of knowledge on the relative importance of other environmental drivers and mechanisms of transmission and dispersion of pathogens in the ocean, and on the spatial scales at which infective stages and host larvae may travel (reviewed in *Groner et al., 2016*; *Shields, 2018*; *Glidden et al., 2022*). In the absence of this information, culling as a potential mechanism to manage the PaV1 disease remains contentious.

During stage B, when culling was conducted, the probability of finding diseased lobsters varied with site, but was significantly higher in only two of the five casita sites relative to stage A, supporting the notion that small-scale habitat and community characteristics (*e.g.*, habitat complexity, types of substrate, species diversity) can play important roles in disease ecology (*Small & Pagenkopp, 2011*; *Lafferty, 2017*; *Davies, Briones-Fourzán & Lozano-Álvarez, 2019*). In the Florida Keys, high variability in disease prevalence in individual sampling sites was also common (*Behringer et al., 2011*; *Butler et al., 2015*). In Bahía de la Ascensión, disease prevalence was consistently higher on more vegetated sites, suggesting that vegetation could act as an environmental reservoir of PaV1 (*Briones-Fourzán et al., 2012*; *Davies, Briones-Fourzán & Lozano-Álvarez, 2019*), as it does for other

pathogens (*Small & Pagenkopp, 2011*). However, in the relatively narrow Puerto Morelos reef lagoon, vegetation is highly dynamic due to hurricanes, herbivore pressure, and inputs of nutrients from anthropogenic sources (*Rodríguez-Martínez et al., 2010*; *van Tussenbroek, 2011*; *Caballero-Aragón et al., 2022*). Also, the possible existence of other animals acting as reservoirs for PaV1 cannot be excluded (*Davies, Briones-Fourzán & Lozano-Álvarez, 2019*; *Davies et al., 2020*).

In the Puerto Morelos reef lagoon, average prevalence of the PaV1 disease increased from 2.5% in 2001 to 10.5% in 2006 (*Lozano-Álvarez et al., 2008*), to 15–20% between 2009 and 2015 (the present study). It further remained around 16–20% between 2016 and 2022, suggesting that it has leveled off (*Davies et al., 2020*; P Briones-Fourzán, pers. obs., June 2021 and June 2022). This also appears to be the case in Bahía de la Ascensión, where the average prevalence remained around 5% between 2008–2010 (*Candia-Zulbarán et al., 2012*) and 2016–2017 (*Davies, Briones-Fourzán & Lozano-Álvarez, 2019*). In both locations, real prevalence is estimated to be at least twice as high (Bahía de la Ascensión: *Huchin-Mian et al., 2013*; Puerto Morelos: *Candia-Zulbarán, Briones-Fourzán & Lozano-Álvarez, 2019*). The higher prevalence levels in the Puerto Morelos reef lagoon reflect that the local population of lobsters consists mostly of juveniles $\leq$ 50 mm CL, which are the most susceptible to PaV1, whereas in Bahía de la Ascensión the local population spans from juveniles to adults.

## CONCLUSIONS

The present study suggests that conservation efforts to enhance juvenile lobsters using scaled-down casitas in shelter-poor habitats is a viable option. The use of casitas did not increase PaV1 prevalence and culling clinically infected lobsters, at least with the periodicity that we used, did not decrease disease prevalence. On the contrary, prevalence was higher throughout the culling period. These results suggest that other factors, such as small-scale habitat and community characteristics, may be influencing disease dynamics. Therefore, upon using casitas to enhance juvenile *P. argus*, previous baseline surveys would be advisable (*Shields, 2018*) as well as monitoring prevalence levels at least once a year during the first few years (*e.g.*, *Davies et al., 2020*).

## ACKNOWLEDGEMENTS

We greatly acknowledge the help in field and laboratory activities of JP Huchin-Mian, IH Segura-García, AF Espinosa-Magaña, R Martínez-Calderón, R Muñoz de Cote-Hernández, L Cid-González, N Luviano-Aparicio, R González-Gómez, and PS Morillo-Velarde. E Escalante-Mancera and MA Gómez-Reali provided technical support, and L Celis-Gutiérrez helped with literature search.

### Funding

This work was supported by Consejo Nacional de Ciencia y Tecnología (CONACYT, México) (project 82724, granted to Enrique Lozano-Álvarez). Rebeca Candia-Zulbarán also received a PhD scholarship from CONACYT. The funders had no role in study design, data collection and analysis, decision to publish, or preparation of the manuscript.

### Grant Disclosures

The following grant information was disclosed by the authors:
Consejo Nacional de Ciencia y Tecnología (CONACYT, México): 82724.
CONACYT.

### Competing Interests

The authors declare there are no competing interests.

### Author Contributions

- Rebeca Candia-Zulbarán conceived and designed the experiments, performed the experiments, analyzed the data, prepared figures and/or tables, authored or reviewed drafts of the article, and approved the final draft.
- Patricia Briones-Fourzán conceived and designed the experiments, analyzed the data, prepared figures and/or tables, authored or reviewed drafts of the article, and approved the final draft.
- Fernando Negrete-Soto conceived and designed the experiments, performed the experiments, authored or reviewed drafts of the article, and approved the final draft.
- Cecilia Barradas-Ortiz conceived and designed the experiments, performed the experiments, prepared figures and/or tables, authored or reviewed drafts of the article, and approved the final draft.
- Enrique Lozano-Álvarez conceived and designed the experiments, analyzed the data, authored or reviewed drafts of the article, and approved the final draft.

### Field Study Permissions

The following information was supplied relating to field study approvals (i.e., approving body and any reference numbers):

Field experiments were approved by Comisión Nacional de Acuacultura y Pesca, México (DGOPA.12019.031108.3134, DGOPA-06695.190612.1737, and PPF/DGOPA-259/14).

### Data Availability

The raw data are available in the Supplemental Files.

### Supplemental Information

Supplemental information for this article can be found online at http://dx.doi.org/10.7717/peerj.15073#supplemental-information.

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
