# Peer review of "Artificial shelters and marine infectious disease: no detectable effect of the use of casitas to enhance juvenile Panulirus argus in shelter-poor habitats on a viral disease dynamics"

_PeerJ, doi:10.7717/peerj.15073_

## Round 0.1 · original submission · Major Revisions

I agree with both reviewers that albeit interesting, the manuscript lacks clarity in most sections, especially in the description of the Methodology used. Do kindly address the concerns and suggestions raised by both reviewers and I look forward to reading the revised version of the manuscript.

Reviewer 1 ·

Basic reporting

Overall the basic reporting metrics were met. There are a few issues i have addressed below:

References:
Herrnkind et al. 2001 and Zarco-Perello & Enriquez 2019 are missing from the citation list.
Genovart et al. 2010, Gittens & Butler 2018, Martin et al. 2006, and Marx & Hernkind 1985 are not cited in the text but appear in the citations list.

Figure 5 the y axis has an & sign rather than a %.

Experimental design

I have some issues with the experimental design and the description of the methods.

- Data analyses appears before the description of stage B, and there is no separate data analyses section for stage B.
-Confusing language/description L263-269. This section is unclear.
-Methods for stage B are not fully developed. The discussion says these sites were visited multiple times, but this is not addressed in the experimental design for stage B. It would be nice to know how many time diseased lobsters were removed, and how many were culled at each sites. To properly understand if culling disease animals has an impact on prevalence, it would be more beneficial to analyze this data with a repeated measures framework. The number of culled diseased animals is important, rather than just the presence of absence of disease, as a concentration of diseased animals can lead to higher transmission rates.
-The reference level for this analysis seems like it should be stage A (without culling) rather than with culling. After culling the rates of disease appear to go up, so it would be good to look at these values over the repeated culling.

Validity of the findings

- Stage B is problematic. I think the authors should reanalyze their findings using a repeated measures design. It does not seem valid to conclude that disease culling does not impact prevalance when the authors state that they may have gone too infrequently. The number of animals removed at each culling event over the two years should be reported so the reader has a better understanding on if this effort was successful.

-I do not see the 0.5 sensitively of visual disease vs subclinical reported in the citations listed on L 449. Did this number come from someplace else? Unpublished data from outside of mexico suggests that subclinical infections can be as much as 8x higher than the visual infections.

Lastly, in order to determine if casitas impact prevalence for juveniles, a full molecular analysis of prevalence should have been conducted. With such high prevalence rates, it is important to understand the amount of subclinical infection also occurring at casitas. Casitas were sampled every 3-4 months to avoid serial sampling, so subclinical infected animals would not likely be caught in the second sampling as they likely moved out of the study area during that time.

Reviewer 2 ·

Basic reporting

no comment

Experimental design

no comment

Validity of the findings

no comment

Additional comments

Comments to the Author:

In this work, the authors concluded that the use of casitas does not increase PaV1 prevalence and that culling clinically infected lobsters, at least with the periodicity that they used, had no measurable impact on disease prevalence. Therefore, they suggested that the casita-based fisheries for P. argus can probably live with PaV1 but should not expect to eradicate it. The study, to some extent, is meaningful and valuable.
However, the manuscript was not written scientifically, the author wrote a lot of background information and useless contents, and the descriptions of some sections were very chaotic and illogical, which seriously affected the quality and readability of the article. Besides, the results obtained and presented in this work were less and cannot fully support the conclusions in this paper. Further, the work was conducted from 2009 to 2015, and I am confused that why not the author published the study previously. I doubt the values and meanings of the results obtained for the present situation.
Therefore, I suggest the manuscript cannot be accepted, at least in its current form. The following suggestions can be noted if you want to resubmit it.

1. The abstract should be accurate and concise, and it is suggested to reduce the words in this section.
2. Line 23: how to define small and large juveniles?
3. There are eight paragraphs in the section of introduction, and it is too many for the present study type, therefore, it is suggested to simplify and delete some redundant and unnecessary descriptions.
4. Line 78, change “some” to “a”.
5. It is unfamiliar to most readers that what is a ‘casita’. So, it would be better to provide a photo of ‘casita’ used in this study in the Materials & Methods section.
6. In the Materials & Methods section, you just need describe the methods you used, and it is not necessary present every detail and aim, which makes it very chaotic, illogical and incomprehensible.
7. The Data analyses is confused and chaotic, please rewrite this section.
8. In Figure 2, the standard errors of control sites were so high, especially in Figure 2C, which makes the science and credibility of the data doubtable.
9. In Figure 5, I doubt that maybe the so high standard errors lead to no significant differences between stage A and B.
10. The quality of figures needs to be improved. What is the meaning of “&” in the Y-axis in Figure 5.
11. You can put Table 1 and Table 2 together. It is not very necessary to present some parameters, e.g., DF, MS, F.
12. Some descriptions in this manuscript were not quite in line with the English expression habits. Please revise the manuscript by a fluent English speaker.

---

## Round 0.2 · Major Revisions

As the recommendations of the reviewers varied greatly, I hereby included some of my comments and recommendation on the manuscript as well.

1. Although the manuscript contains valid results, I would have to agree with the reviewers that the current flow of the manuscript, especially the Methodology section, warrants some revisions.

2. Line 445-450: the explanation of Experiment Stage B is rather confusing. The authors conducted 'eight additional samplings on casitas sites', and I presumed they (partially) removed diseased lobsters, and revisited the sites? If so, what were the revisit intervals and frequency? Please describe Experiment Stage B, both at line 349-361 and 445-450 clearly.

3. Line 541: Do you mean '442 were found to be diseased and culled'?

4. Back to Experiment Stage B, if (at lines 359-360) the authors removed all diseased lobsters at casitas sites, what would be the control then?

5. Additionally, Line 539-773 (the Results of Experiment Stage B), the authors are comparing stage A and stage B, but if I understand correctly, Experiment Stage B is a separate experiment on its own, aimed 'to examine whether culling all diseased lobsters on each sampling date altered further overall prevalence levels and whether such changes were consistent among sites' (line 350-352). The current Results section for Experiment Stage B does not address this research question. Ideally, in Stage B, the authors should have had several casitas with infected lobsters being culled/removed, and several with known infected lobsters but not removed. The two treatments are then revisited to account for whether culling might affect/alter the prevalence level.

6. Line 687-689: If 'disease prevalence tends to be higher in Stage B' (line 542, 571-572), then how would 'culling more often or more intensely ... may help mitigate the effects of this disease'? This suggestion is contradictory to the results.

Reviewer 1 ·

Basic reporting

All previous issues have been addressed

Experimental design

The new revisions greatly increase the reproducibility of this experiment.

Validity of the findings

The added information to the discussion have greatly benefited this paper.

Reviewer 2 ·

Basic reporting

NO

Experimental design

NO

Validity of the findings

NO

Additional comments

Even through the mauscript has been thoroughly revised, I also confued with this study. First, I dont think the authors have reviesd this manuscript carefully. Besides, even the 95% confidence intervals were uesd in the manuscript, the data in Figure 3 and 5 were also doubtable. Therefore. I dont think this mannscript should be accepted.

Reviewer 3 ·

Basic reporting

This study shows no relationship between casitas usage with PaV1 disease prevalence in lobsters. It is well-written and clear. But, I suggest using the latest references throughout the manuscript as current references are pretty outdated.

Experimental design

The experimental design is straightforward, except for Line 296 – 300 (Data analyses for Experimental Stage B). I suggest adding more details on this method.

Validity of the findings

The authors used statistical tests to analyze the results obtained. The conclusion is well-stated.

Additional comments

Please use updated references.
There are some sentences that can be improved, e.g. Line 227 – 229, Line 296 – 300, etc.
Suggestion: Elaborate more on the methods of Data analyses- Experimental stage B.

---

## Round 0.3 · Minor Revisions

There is just a minor comment from the reviewer on the missing scale bar from Figure 2.

Reviewer 3 ·

Basic reporting

Sufficient background and literature references are relevant.

Experimental design

The experimental design is clear

Validity of the findings

Results, discussion and conclusions are well stated.

Additional comments

Authors have clearly addressed all previous comments.

Reviewer 4 ·

Basic reporting

The language of the manuscript is clear and easy to follow. The overall flow of the whole manuscript is smooth, and sufficient literature was used to support, both the introduction and discussion. I noticed that this is the third round of revision of the manuscript, and based on the rebuttal letter, I would agree to the revisions done by the authors.

Just a minor comment, could the authors add a scale bar to Figure 2? This will facilitate readers in gauging the size of the casitas and lobsters.

Experimental design

no comment.

Validity of the findings

Replications are valid and the statistical analyses conducted are appropriate.

Additional comments

I would say this is a very interesting on-site experimental design that allows direct observation of lobsters in their natural habitat and the effect of casitas usage on viral disease dynamics.

---

## Round 0.4 · accepted · Accept

Thank you for addressing all the concerns and suggestions from the reviewers. Im looking forward to reading the published version!